# Different HCV Exposure Drives Specific miRNA Profile in PBMCs of HIV Patients

**DOI:** 10.3390/biomedicines9111627

**Published:** 2021-11-05

**Authors:** Daniel Valle-Millares, Óscar Brochado-Kith, Luz Martín-Carbonero, Lourdes Domínguez-Domínguez, Pablo Ryan, Ignacio De los Santos, Sara De la Fuente, Juan M. Castro, María Lagarde, Guillermo Cuevas, Mario Mayoral-Muñoz, Mariano Matarranz, Victorino Díez, Alicia Gómez-Sanz, Paula Martínez-Román, Celia Crespo-Bermejo, Claudia Palladino, María Muñoz-Muñoz, María A. Jiménez-Sousa, Salvador Resino, Verónica Briz, Amanda Fernández-Rodríguez, on Behalf of Multidisciplinary Group of Viral Coinfection HIV/Hepatitis (COVIHEP)

**Affiliations:** 1Unit of Viral Infection and Immunity, National Center for Microbiology, Institute of Health Carlos III, 28220 Majadahonda, Madrid, Spain; da.valle@isciii.es (D.V.-M.); obrochado@isciii.es (Ó.B.-K.); agsmm73@hotmail.com (A.G.-S.); paula.mroman@isciii.es (P.M.-R.); jimenezsousa@isciii.es (M.A.J.-S.); sresino@isciii.es (S.R.); 2Hospital La Paz Institute for Health Research (IdiPAZ), 28046 Madrid, Spain; lmcarbonero@gmail.com (L.M.-C.); juanmi.castro@gmail.com (J.M.C.); mariocoslada@gmail.com (M.M.-M.); 3VIH Servicio de Medicina Interna Research Institute Hospital 12 de Octubre (i+12), 28041 Madrid, Spain; lourdes.dd@outlook.com (L.D.-D.); estudioshiv12@gmail.com (M.L.); marmatamo@gmail.com (M.M.); 4Department of Infectious Diseases, Infanta Leonor Teaching Hospital, 28031 Madrid, Spain; pabloryan@gmail.com (P.R.); guillermocuevastascon@gmail.com (G.C.); victorino.diez@gmail.com (V.D.); 5Internal Medicine Servicie Hospital Universitario de La Princesa, 28006 Madrid, Spain; isantosg@salud.madrid.org; 6Internal Medicine Service Hospital Puerta de Hierro, 28222 Madrid, Spain; sarafm28@hotmail.com; 7Laboratory of Reference and Research on Viral Hepatitis, National Center for Microbiology, Institute of Health Carlos III, 28220 Majadahonda, Madrid, Spain; ccrespo@externos.isciii.es (C.C.-B.); veronica.briz@isciii.es (V.B.); 8Research Institute for Medicines (iMed.ULisboa), Faculty of Pharmacy, Universidade de Lisboa, 1649-003 Lisbon, Portugal; palladino.claudia@gmail.com; 9Department of Animal Genetics, Instituto Nacional de Investigación y Alimentación Agraria y Alimentaria (INIA), 28040 Madrid, Spain; mariamm@inia.es; 10Faculty of Medicine, Universidad Alfonso X el Sabio, Avenida Universidad 1, 28691 Villanueva de la Cañada, Madrid, Spain

**Keywords:** HCV, HIV, microRNAs, spontaneous viral clearance, high throughput sequence

## Abstract

Micro RNAs (miRNAs) are essential players in HIV and HCV infections, as both viruses modulate cellular miRNAs and interact with the miRNA-mediated host response. We aim to analyze the miRNA profile of HIV patients with different exposure to HCV to explore specific signatures in the miRNA profile of PBMCs for each type of infection. We massively sequenced small RNAs of PBMCs from 117 HIV+ infected patients: 45 HIV+ patients chronically infected with HCV (HIV/HCV+), 36 HIV+ that spontaneously clarified HCV after acute infection (HIV/HCV-) and 36 HIV+ patients without previous HCV infection (HIV). Thirty-two healthy patients were used as healthy controls (HC). Differential expression analysis showed significantly differentially expressed (SDE) miRNAs in HIV/HCV+ (*n* = 153), HIV/HCV- (*n* = 169) and HIV (*n* = 153) patients. We found putative dysregulated pathways, such as infectious-related and PI3K signaling pathways, common in all contrasts. Specifically, putatively targeted genes involved in antifolate resistance (HIV/HV+), cancer-related pathways (HIV/HCV-) and HIF-signaling (HIV) were identified, among others. Our findings revealed that HCV strongly influences the expression profile of PBMCs from HIV patients through the disruption of its miRNome. Thus, different HCV exposure can be identified by specific miRNA signatures in PBMCs.

## 1. Introduction

Approximately 2.75 million individuals worldwide are simultaneously coinfected with hepatitis C virus (HCV) and HIV, as both share common transmission routes (sexual and injection drug use (IDU) [1]. HIV has a negative impact on the progression of HCV infection, and their natural history is altered in coinfection with HCV leading to a higher rate of cirrhosis, liver failure, and hepatocellular carcinoma, among others [2].

HCV causes acute and chronic infections. Acute infection is usually asymptomatic and 20–40% of infected persons spontaneously clear the virus within six months without treatment [3]. A clearance occurs more frequently in young people and women [4], besides a certain genetic predisposition, as people of African origin are less likely to clear HCV than those of Caucasian origin, although the reasons for these differences are unclear.

HIV/HCV coinfection hinders spontaneous HCV clearance [5], with the frequency of spontaneous clearance in coinfected individuals lower (between 5–10%) than monoinfected patients (12–50%) [6]. Spontaneous HCV clearance can be predicted by the presence of factors, such as high viral load, or the presence of certain alleles in the single nucleotide polymorphisms (SNPs) rs12979860 [7,8] and rs8099917 [9] located at the *IL28B* gene encoding interleukin 28, also known as interferon (IFN) lambda type III. The rs12979860 CC genotype has been associated with a higher frequency of spontaneous HCV resolution. Additionally, spontaneous clarification of HCV is reflected in the transcriptional miRNA profile of PBMCs, as it has been shown in HCV monoinfected patients [10].

MicroRNAs (miRNAs) are a type of non-coding RNAs that play an essential role in the regulation of gene expression. These regulatory molecules have been shown to be involved in the HIV and HCV infectious cycle, also regulating innate and adaptive immune responses against viral infections. These miRNAs are susceptible to molecular alterations due to viral infections, providing information on infectious dynamics and virus-host relationships, as well as the consequences of the immune response to infection [11].

Despite hepatocytes being the target cells for HCV infection, peripheral blood mononuclear cells (PBMCs) constitute its main extrahepatic reservoir, being able to co-exist with HIV in coinfection. PBMCs miRNA profile changes provide us information on innate and adaptive immune responses against HIV and HCV infections or even viral hijacking. For instance, the hsa-miR-122-5p, a liver-specific miRNA that modulates the liver homeostasis and its HCV hijacking to favor the HCV viral cycle [12], while the hsa-miR-223 promotes HIV latency in CD4+ T cells of HIV-1 patients by binding to the viral 3′ UTR [13]. Many questions remain unanswered in miRNA profile characterization, especially in the context of HIV/HCV coinfection, where scarce data have been published and none with a massive approach, which may be an additional key to elucidate virus-host relationships and their consequences.

Thus, our study aims to massively characterize the miRNA profile alterations due to chronic HCV infection or acute infection followed by HCV spontaneous elimination in PBMCs of HIV patients. In addition, this study assesses specific miRNA signatures in PBMCs entirely due to each infection, as well as the potential consequences of these alterations at the molecular function level.

## 2. Materials and Methods

This cross-sectional study was conducted from 2016 to 2017. Patients were recruited from five Public Spanish Hospitals in Madrid Autonomous Community: Hospital Universitario 12 de Octubre, Hospital Universitario La Paz, Hospital Universitario Infanta Leonor (Madrid), Hospital Universitario La Princesa and Hospital Puerta de Hierro (see Appendix A). Samples were processed at the National Center for Microbiology, Institute of Health Carlos III, Madrid (Spain).

### 2.1. Patient Groups

A total of 149 patients, 117 HIV+ and 32 healthy controls (HC), all of them Europeans of Caucasian origin, were enrolled in this study. Patients and controls were age and gender matched. All HIV+ patients were receiving suppressive antiretroviral treatment (ART) for at least one year and had CD4+ T-cells counts ≥ 500 cel/mm^3^ for at least one year before sample collection. HIV+ patients were classified into three groups: (1) HIV+/HCV+ group (*n* = 45): HIV patients with active HCV-chronic infection naïve to any HCV treatment (positive PCR and positive HCV antibodies); (2) HIV+/HCV- group (*n* = 36): HIV+ patients who had been exposed to HCV but experienced spontaneous viral clearance during the first 6 months after HCV infection (negative PCR and positive HCV antibodies); (3) HIV+ group (*n* = 36): HIV monoinfected patients that had never been infected with HCV (negative PCR, negative HCV antibody and no previous HCV treatment).

### 2.2. High Throughput Sequencing of Small RNA

Total RNA, including miRNAs, was isolated from PBMCs within the first 4 h after blood extraction with the miRNeasy Mini kit (Qiagen, Germantown, MD, USA). Small RNA library synthesis was constructed with TruSeq Small RNA kit v.4 (Illumina, San Diego, CA, USA) and sequencing were performed at the Centre for Genomic Regulation (CRG) in Barcelona (Spain), as previously described [10].

### 2.3. Data Processing Pipeline

The miRNA sequencing data were processed and analyzed following a miRNA-specific bioinformatics pipeline for the quality control, alignment, quantification and identification of host miRNA sequences (see Appendix A for further information).

### 2.4. Statistical Analysis

Clinic and demographic data were summarized as medians for continuous variables and frequency and percentage for categorical variables. Significant differences were calculated using the Kruskal–Wallis and Mann–Whitney U test for continuous variables and the Fisher’s exact test for categorical variables.

#### 2.4.1. Data Preprocessing

Firstly, we used the *filterByExpr* function to retain those miRNAs that have sufficiently large counts according to the minimum expression criterion [14]. Second, we accounted for sequencing depth normalization with the trimmed mean of M-values normalization method (TMM) and the count per million (CPM) approach.

#### 2.4.2. Data Exploratory Analysis

Partial least square discriminant analysis (PLS-DA) with the normalized miRNA expression profile was performed to evaluate the baseline performance of discriminant methods.

#### 2.4.3. Differential Expression Analysis

Significant differentially expressed (SDE) miRNAs were obtained by a negative binomial generalized linear regression model (NB-GLM) with the R package MASS v7.3-53 [15], where:(1)$$Y_{ij}=T_{j}+e_{ij}$$

*Y* is the normalized count value for the *i_th_* miRNA in the dataset, *T* is the model’s explanatory variable with levels *j* = 1 for HC and *j* = 2 for HIV, HIV/HCV+ or HIV/HCV- groups, for each comparison and *e* is the random residual term. SDE miRNAs were identified by a statistically significant *p*-value < 0.05 adjusted by false discovery rate (FDR) using Benjamin–Hochberg correction and a fold change (FC) higher than 1.5 (LogFC > 0.585).

We used the R software for statistical computing (v4.0.3) (www.r-project.org) for the statistical analysis.

### 2.5. miRNA-Based Target Prediction and Pathway Enrichment Analysis of Target Genes

We explored target genes and potential altered molecular pathways for the SDE miRNAs identified in the statistical analysis.

The miRTarbase database was used to identify experimentally validated interactions throughout the hypergeometric distribution approach. Statistically significant miRNA gene-targets (enriched target genes) were found at *p*-value < 0.05 adjusted by FDR using Benjamin–Hochberg correction and a minimum of two interactions.

The g:Profiler web-based server was used to conduct a functional enrichment analysis on the list of significantly enriched gene targets [16]. We choose the KEGG (Kyoto Encyclopaedia of Genes and Genomes) database to retrieve statistically significantly enriched pathways at the adjusted *p*-value < 0.05. The Gene Ontology (GO) annotation database was used as a source of information for the enriched target genes identified by the SDE miRNA uniquely found in each comparison.

## 3. Results

### 3.1. Clinical Characteristics of Each Group of Patients

Table 1 shows the epidemiological and clinical characteristics of each group of patients, HIV+, HIV/HCV+, HIV/HCV- and HC. We only found significant differences for the HIV transmission route (*p* < 0.001) and the genotype for rs12979860 polymorphism at the *IFNL4* gene, where HIV+/HCV- patients showed a higher proportion of the favorable CC genotype (*p* = 0.001), as expected. There were no differences in CD4+ or CD8+ T-cell parameters between any HIV+ patients.

We also explored the metabolic characteristics of the patients (Table 2), and we found differences regarding weight (*p* = 0.015) and body mass index (*p* = 0.026), where HIV/HCV+ coinfected patients displayed lower levels. Regarding the liver profile parameters, we found significant differences compared to HC in: total cholesterol (TC), high-density lipoprotein (HDL) and atherogenic index of plasma (AIP) for patients in the HIV/HCV+ group; HDL, triglycerides (TG), atherogenic index (AI) and AIP in HIV/HCV- patients; HDL, AI and AIP in HIV patients. Additionally, liver transaminases displayed significantly higher levels in HIV/HCV+ coinfected patients relative to the HC group and to the rest of the HIV-1 infected patients.

### 3.2. Differentially Expressed miRNAs between Study Groups

The raw sequencing data of the HIV-1 infected patients can be accessed at the ArrayExpress repository (EMBL-EBI) under the accession number E-MTAB-8274 for HIV patients and at E-MTAB-8023 for the HC individuals.

On average, 10.9 million reads per sample were obtained, which is an appropriate depth for expression analysis. A total of 54% of the reads were mapped to the reference genome, identifying 85.3% of the miRNAs in miRBase. A total of 1662 miRNAs were identified; those with no informative expression were filtered out on each comparison. Figure 1 shows a summary of the study and the significant differences identified. 

The PLS-DA of normalized, log-transformed and scaled miRNA expression data show us a clear difference of HC control group with respect to the rest of the study groups, whilst HIV samples are all grouped together (Appendix A).

#### 3.2.1. HIV/HCV+ Co-Infected Patients

Overall, 535 miRNAs surpassed the filter of minimum expression criteria. Among them, 153 SDE miRNAs were identified in HIV/HCV+ patients compared to HC. After assessing the top expression changes (Log_2_FC < −2 or Log_2_FC > 2) we identified that the hsa-miR-1248, hsa-miR-3960, hsa-miR-4508 and hsa-miR-10401-3p were strongly downregulated in HIV/HCV+ patients (Figure 2A-left and Appendix A).

The target enrichment analysis showed that these 153 miRNAs could modulate the expression of 66 genes (Figure 2B, dark blue; Appendix A), which were mainly involved in infectious-related pathways (toxoplasmosis, legionellosis and EBV), antifolate resistance and the PI3K-Akt signaling pathway (Figure 2C, purple; Appendix A). Further, the vast majority of enriched genes identified in this comparison were not shared with the HIV/HCV- spontaneous clearance and HIV analyses.

#### 3.2.2. HIV/HCV- Spontaneous Clearance

We analyzed a total of 509 miRNAs that fulfilled the minimum expression criterion, identifying 169 SDE miRNAs in the HIV/HCV- group with respect to HC. Moreover, the hsa-miR-1246, hsa-miR-1248, hsa-miR-3960 and hsa-miR-4508 had the strongest downregulation (Log_2_FC < −2) (Figure 2A-middle and Appendix A).

We explored the potential alteration at the molecular level, and we found a set of 27 enriched genes whose expression could be modulated by the 169 SDE miRNAs in HIV/HCV- patients (Figure 2B, pink; Appendix A). These genes are significantly involved in 11 functional pathways. Among them, several infection-related pathways were identified, such as HIV-1, toxoplasmosis, hepatitis B and Epstein–Barr. Additionally, the insulin-signaling pathway, miRNAs in cancer, chemokine and PI3K-AKt signaling pathways were also represented, among others (Figure 2C, pink; Appendix A).

#### 3.2.3. HIV+ Infected Group

We identified 153 SDE miRNAs (out of 521 analyzed) in the HIV+ group with respect to the HC. As observed before, we found that the strongest alterations in the miRNA profile of HIV+ patients occurred for the hsa-miR-1246, hsa-miR-1248, hsa-miR-3960 and hsa-miR-4508 (Log_2_FC < −2) (Figure 2A-right, Appendix A).

The miRNA-gene interaction analysis showed 15 enriched target genes for the SDE miRNAs in the HIV+ group (Figure 2B, light blue; Appendix A). According to the pathway enrichment analysis, these genes are significantly enriched in seven functional pathways (Figure 2C, blue; Appendix A) being the most significant the hypoxia-inducible factor 1 (HIF-1) signaling pathway, and five related to different infections such as the Epstein–Bar virus, the HIV, Chagas diseases toxoplasmosis and measles. The AGE-RAGE signaling pathway in diabetic complications was also identified.

#### 3.2.4. Specific Viral Infection Group Signatures Due to HIV+, HIV/HCV+ or HIV/HCV-

Despite 109 differentially expressed miRNAs commonly found in all the previous analyses, hsa-miR-33, hsa-miR-122-5p and hsa-miR-125a-5p were the most relevant shared miRNAs, which indicates the specific miRNA profile characteristic of HIV infection, we focused on the non-shared SDE miRNAs to identify potential specific fingerprints for each group of patients (HIV/HCV+, HIV/HCV- or HIV). Thus, we selected the unique SDE miRNAs across each comparison with the HC group (N°_SDE miRNAs_ HIV/HCV+: 25; N°_SDE miRNAs_ HIV/HCV-: 24; N°_SDE miRNAs_ HIV+: 13) (Figure 3A).

The analysis of the miRNA-target gene for each specific group of patients showed us specific interactions that were further analyzed by up and downregulated miRNAs separately (Figure 3B,C):

HIV/HCV+ group: We explored the putative target genes for the 25 unique SDE miRNAs. Among the top 10 target genes, we found enriched the Spectrin Beta Non-Erythrocytic 1 (*SPTBN1*) gene, targeted by the downregulated hsa-miR-589-3p, hsa-miR-423-3p and hsa-miR-331-3p (Appendix A). The *SPTBN1* gene plays a relevant role in biological processes related to the regulation of SMAD proteins and cytoskeleton movement guider. Additionally, the miRNA-target analysis revealed that autophagy-related 5( ATG5), protein kinase AMP-activated catalytic subunit alpha 1(*PRKAA1*), *MYCN*, integrin subunit beta 3 binding protein (*ITGB3BP*) and platelet-derived growth factor receptor beta (*PDGFRbeta*) might be potential targets for upregulated hsa-miR-582, hsa-miR-299, hsa-miR-34a and hsa-miR-9 in HIV/HCV+ patients.

HIV/HCV- group: From the unique 24 SDE miRNAs in HIV/HCV- patients, we found the *abraxas 2*, BRISC complex subunit (*ABRAXAS2*) gene, targeted by: downregulated hsa-miR-629-3p and hsa-miR-627-3p (Appendix A). Due to its polyubiquitin modification function, *ABRAXAS2* plays a key role in protein deubiquitination, cell division and chromosome segregation, among others. Moreover, miRNA-target prediction of upregulated miRNAs showed significantly enriched target genes, such as claudin 1(*CLDN1*), that could be repressed by hsa-miR-376c-3p, hsa-miR-1304-3p and hsa-miR-337-3p in PBMCs of HIV/HCV- patients.

HIV+ group: The 13 unique SDE miRNAs showed a panel of 10 genes that interact with downregulated hsa-miR-4516, hsa-miR-542-3p and hsa-miR-494-3p (Appendix A). Among them, the zinc-binding alcohol dehydrogenase domain containing 2(*ZADH2*) encodes for an oxidoreductase that negatively regulates fat cell differentiation.

## 4. Discussion

We have elucidated for the first time that the PBMCs miRNA profile of HIV patients is specifically dysregulated according to the HCV exposure, being different for chronic hepatitis C (CHC) infection, acute infection followed by spontaneous clearance or those never infected by HCV. Moreover, here we report potential miRNAs and target genes whose expression may be compromising, and thus the biological processes in which they are involved.

To date, only scarce data have been published regarding the miRNA profile of HIV/HCV patients [17,18], none of them with a massive approach and even less focused on spontaneous clarifiers.

Analysis of clinic and metabolic characteristics showed significant deregulated levels of lipid parameters of the three groups of HIV patients with respect to HC, while no differences were found within HIV patients. The AIP, which is an index of dyslipidaemia that accurately correlates with the size of HDL-C and LDL-C particles [19], was higher for all HIV groups, especially for HIV/HCV+. Accordingly, we found dysregulation of key lipid metabolism miRNAs, such as hsa-miR-33a, hsa-miR-122-5p and hsa-miR-125a-5p [20,21] that were downregulated in all HIV patients. Both HIV and HCV infections disturb host lipid metabolism generating abnormalities that could lead to severe pathologies [22]. Thus, the miRNA profile of PBMCs reflects common comorbidities in HIV/HCV patients regarding lipid metabolism, such as an increased cardiometabolic risk, dyslipidaemia, insulin resistance and hepatic steatosis [23].

Furthermore, all HIV patients showed a strong downregulation of miR-1248, miR-4508 and miR-3960. This feature gives us a common pattern between HIV groups highlighting a shared signature of miRNA profile deregulation, probably due to immune response against viral infections. Previously, the miR-1248 was positively correlated with immune system-dependent processes, such as inflammation [24]. Thus far, little is known about these miRNAs, and their relevance in HIV infection needs to be further explored.

### 4.1. HIV/HCV+ miRNA Profile

HIV/HCV coinfection leaves a fingerprint of 153 dysregulated miRNAs in PBMCs. We found that these miRNAs might modulate the expression of genes involved in the PI3K-Akt signaling pathway, which have been previously associated with the HCV infection and immune response against other pathogens [25]. Among these putative miRNAs, we highlight the miR-126-5p, hsa-miR-34a-5p and hsa-miR-148b-3p, whose expression levels were significantly higher in HIV/HCV coinfected PBMCs compared to HC. The miR-126-5p hepatic expression has been associated with HCV viral load [26], the circulating hsa-miR-34a-5p promotes liver fibrosis in CHC [27], and the hsa-miR-148b-3p has been previously identified as dysregulated in PBMCs of HIV patients [28]. Although these miRNAs were also associated with the regulation of carcinogenesis and antitumor responses [29,30], we found them involved in the PI3K-Akt signaling pathway. Both HIV and HCV infections promote the activation of the PI3K-Akt signaling pathway, often resulting in other pathogenesis, such as increasing the risk of viral carcinogenesis, and triggering immune-dependent processes, such as inflammation [31,32]. Therefore, the enhanced expression that we have observed of miR-126-5p, miR-34a-5p and miR-148b-3p in HIV/HCV patients—and their putative target genes *HOTAIR*, *SLC45A3*, *UGT8*, *VEGFA*—could be associated with a higher immune pro-inflammatory responses and deregulation of carcinogenic genes [30,33], which is common in HIV/HCV patients.

Additionally, we also identified the downregulation of the well-known miR-125a/b-3p and miR-122-5p in HIV/HCV+ patients, which were also previously observed in the PBMCS of HCV chronic patients and HIV/HCV- patients, compared to HC [10]. The miR-122 is a specific and highly abundant liver miRNA that favors HCV replication in hepatocytes and other non-hepatic target cells due to HCV viral hijacking [34]. According to our data set, miR-122-5p is downregulated in HIV/HCV coinfection and modulates the expression of *COPA* and *OSBP*. Both genes are positively associated with the mechanism of replication of HCV within their target cells through the Golgi apparatus mechanism allowing virogenic organelle formation and HCV release [35]. Therefore, miR-122-5p downregulation might allow *COPA* and *OSBP* expression in HIV/HCV patients, promoting HCV replication in PBMCs. In line with our results, Moghoofei et al. 2021 also reported that hsa-miR-122-5p was differentially expressed in PBMCs of both HIV, HCV and HIV/HCV patients [17]. These outcomes indicate that both HCV and HIV influence the expression of hsa-miR-122-5p not only in liver cells but also in PBMCs.

On the one hand, miR-125a-5p has a dual role in blocking the expression of genes involved in the lipid uptake and modulating anti-inflammatory processes [36], while miR-125b-5p is mostly related to lipogenesis by targeting *SCD-1* [37]. Therefore, hsa-miR-125a/b-5p downregulation in the PBMCs of HIV/HCV+ may contribute to excessive lipid accumulation, highlighting a worse prognosis of HIV/HCV patients.

On the other hand, we explored the specific HIV/HCV+ signature of PBMCs. Functional analysis for up and downregulated miRNAs were separately performed to maximize the identification of regulatory events related to phenotypic differences among a group of patients [38]. The upregulation of hsa-miR-34a, hsa-miR-9, hsa-miR-299 and hsa-miR-582 suggest that they may be involved in the suppression of key target genes associated with HCV-host interactions as the *ATG5*, whose inhibition may help to block the HCV replication [39]. Upregulated miRNAs potentially target genes related to liver-disease events, such as *PRKAA1*, *MYCN* and *ITGB3BP* [40,41]. Interestingly, we also observed a potential target of the *PDGFRbeta*, highly expressed by hepatic stellate cells (HSCs). The *PDGFRbeta* is a key mediator in liver injury and fibrogenesis, contributing to human cirrhosis [42]. Therefore, PBMCs, similar to HSCs, are susceptible to suffer alterations at the transcriptome level in response to HCV infection. Thus, although most miRNAs involved in HCV infection have been characterized in hepatocytes, in this study, we showed dysregulated miRNAs in HIV patients chronically infected by HCV involved in liver-related diseases, thereby indicating a similar response between PBMCs and hepatic cells against HCV infection.

Similarly, we also identified specific miRNAs exclusively downregulated in HIV/HCV+ patients (hsa-miR-589-3p, hsa-miR-423-3p and hsa-miR-331-3p), all of them being putative modulators of *SPTBN1* expression. This gene is involved in the TGF-β pathway, among others, where it interacts with *SMAD3*/*SMAD4*. Within the liver, the loss or downregulation of *SPTBN1* expression promotes hepatocellular carcinoma—and other tumor progression—by interrupting the TGF-β pathway and enhancing the Wnt signaling pathway in mice [43] and humans [44]. Since both liver cells and PBMCs are host cells for HCV, they might converge in the dysregulation of miRNA, gene or functional response [45].

### 4.2. HIV/HCV- Spontaneous Clearance Profile

We explored the long-term effects of HCV spontaneous clarification in PBMCs of HIV patients, where 169 SDE miRNAs were observed. The top downregulated miRNAs (hsa-miR-3960, hsa-miR-1248 and hsa-miR-4508) were previously identified in the HIV/HCV+ group. Additionally, we found that the expression levels of hsa-miR-1246 were also lower in HIV/HCV- patients compared to healthy individuals. Hsa-miR-1246 upregulation in serum has been considered a potential biomarker for the early detection of several cancers, such as hepatocellular carcinoma, common in chronic HCV patients [46]. Hsa-miR-1246 was also deeply downregulated in the PBMCs of HIV/HCV- patients, as previously observed by our group in PBMCs after spontaneous clarification of HCV in HCV-monoinfected patients [10]. The downregulation of hsa-miR-1246 may be due to HIV infection [47], but further studies are required to decipher whether a clear association between the spontaneous resolution of HCV and deregulation of hsa-miR-1246 exists.

Despite more cancer-related diseases being reported during chronic HCV infection, our results indicate that several cancer-related routes are highly represented for the 169 SDE miRNAs identified in HIV/HCV- patients. Among these, we report 21 deregulated miRNAs in HIV/HCV- patients as potential targets of the *Programmed Cell Death Ligand 1* (*PD-L1*) pathway in cancer. The *PD-L1* (and its receptor PD-1) is an immune inhibitory factor that blocks cytokine secretion, and therefore, tumor cell apoptosis is the leading cause for cancer escape. The blockage of PD-L1 and PD-1 is closely associated with immune activation and HCV viremia reduction [48]. Therefore, our results indicate that miRNA profile dysregulation highlights the molecular consequences due to HCV resolution, although further investigations are needed to decipher the long-term consequences of HCV spontaneous clearance and cancer-related events.

By exploring the specific footprint of HIV/HCV- patients, we exclusively found 24 SDE miRNAs. Regarding the upregulated miRNAs, we observed that hsa-miR-376c-3p, hsa-miR-1304-3p and hsa-miR-337-3p are putative regulators of *CLDN1*. This gene encodes for a key cellular co-receptor for HCV entry, which increases susceptibility to HCV infection in the target cells where it is expressed [49]. The upregulation of the mentioned miRNAs in HIV/HCV- patients will suppress *CLDN1* expression, and thus, reduce PBMCs’ susceptibility to HCV infection.

On the other hand, we also analyzed those miRNAs specifically downregulated in HIV/HCV- patients. We observed a specific downregulation of hsa-miR-627-3p and hsa-miR-629-3p. These miRNAs are putative targets of *ABRAXAS2*, which is a part of the deubiquitinating enzyme complex BRISC, whose deficiency resulted in the strong attenuation of IFN response [50], which is stimulated by HCV infection [51]. The IFN-signaling pathway has an essential role during the early stages of HCV infection by enhancing and regulating the antiviral immune response [52], where hundreds of genes are activated. Thus, hsa-miR-627-3p and hsa-miR-629-3p may be potential biomarkers that highlight molecular alterations specifically due to HCV resolution in HIV-1 patients.

In addition, roughly 75% of HIV/HCV- patients enrolled in this study presented the favorable interleukin-28B polymorphism (rs12979860) (CC genotype) at *IFNL4,* which is part of type 1 IFNs that exhibit antiviral activity since it has been associated with a high rate of HCV spontaneous resolution [7].

### 4.3. HIV miRNA Profile

Finally, we explored the miRNA expression profile of the HIV group without previous HCV infection, where a total of 153 miRNAs were found dysregulated. Thus, the miRNA profile in PBMCs of HIV patients lacks normalization, thereby indicating persistent host immune activation and molecular pathway alterations in cellular metabolism, several pathways related to host immune response against several pathogens stand out, such as HIV, EBV, Chagas, toxoplasma and measles [53]. Although ART therapy has enormously improved HIV suppression, it fails to eradicate HIV latently in target cells [54]. The HIF-1 signaling pathway was the most significant targeted pathway by the downregulated of hsa-miR-143-3p (putative gene targets: *BCL2*, *SERPINE1*, *HK2* and *AKT1*) among others [55]. The HIF-1 pathway regulates the cellular response to low oxygen through HIFs that control the expression of a wide range of genes involved in energy metabolism and inflammation [56]. Hypoxia directly influences viral proliferation, promoting or suppressing infectivity depending on the virus [57]. Similarly, Santanu et al. 2020 also reported an alteration of the HIF-1 signaling pathway in the PBMCs of HIV-1 patients [28].

Moreover, we explored those specific miRNAs exclusively altered in HIV infection compared to HC. Among the 13 HIV-specific upregulated miRNAs, we found the characteristic miR-223. This microRNA is an important inhibitor of viral replication since it binds to the 3’ UTR end of HIV RNA, inducing viral latency [13]. Thus, the suppression of this miRNA increases the susceptibility of mononuclear cells to HIV-1 infection [13].

Additionally, we explored the specific downregulated miRNAs. Our results showed a panel of three downregulated miRNAs (hsa-miR-4516, hsa-miR-494-3p and hsa-miR-542-3p) involved in the regulation of genes associated with viral latency (*SUPT16H*, *TFPI*) [58], HIV reactivation (*PIM1*) [59] and persistent metabolic and immune disorders (*ZADH2*, *CCL16*). Taken all together, SDE miRNAs in suppressed HIV patients may indicate ongoing abnormalities in PBMCs of HIV suppressed patients and potential biomarkers for HIV latency. With respect to fatty acid metabolism, we also observed that the downregulated miRNAs hsa-miR-494-3p and hsa-miR-4516 also targeted the *ZADH2* gene, a negative regulator of fat cell differentiation. In this setting, HIV-infected patients showed significant alterations of lipid profile (HDL, AI and AIP) and liver transaminases with respect to non-infected patients. Even though HIV-infected individuals on ART reduced AIDS-associated morbidities, one might expect a complete recovery of the metabolic equilibrium; however, HIV-1 latency induces ongoing abnormalities in the lipid profile giving rise to non-AIDS-associated comorbidities [60].

In conclusion, our study offers key insights into the molecular mechanisms of pathogenesis of HCV infection in HIV patients, as well as the opportunity to define a characteristic molecular fingerprint that can be a useful clinical tool for personalized patient management and treatment.

To summarize, roughly 75% (109 SDE miRNAs) of deregulated miRNAs were shared among the different HIV-1 groups (HIV/HCV+, HIV/HCV- and HIV+ vs. HC). HIV/HCV+ showed higher deregulation of infection and cancer-related pathways, HIV/HCV- mainly displayed alteration of cancer-related pathways and HIV+ showed a lack of normalization with remained dysregulation of infection-related pathways, such as HIV and HIF-signaling, among others. Moreover, our findings show a specific miRNA profile dysregulation in the PBMCs of HIV+, HIV/HCV+ and HIV patients after HCV spontaneous clearance. All this together indicates that miRNAs are promising targets for biomarker analysis and may also be a basic pillar of host immune response against viral infections by targeting and regulating protein-coding genes expression.

## Figures and Tables

**Figure 1 biomedicines-09-01627-f001:**
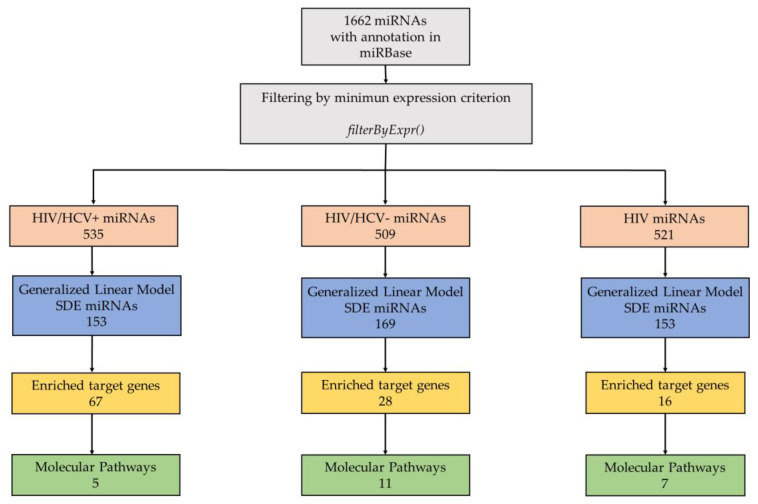
miRNA sequencing analysis and summary of results. Schematic representation of miRNA data analysis and results with respect to the healthy control group. The top grey square shows the total PBMC’s miRNAs in our raw expression matrix with annotations in miRBase. Minimum expression criterion was used to filter out miRNAs with no informative expression. The resulting miRNAs were subject to differential expression analysis (orange). Differentially expressed microRNAs of PBMCs in the HIV groups, their target genes and molecular pathways are represented by blue, yellow and green, respectively.

**Figure 2 biomedicines-09-01627-f002:**
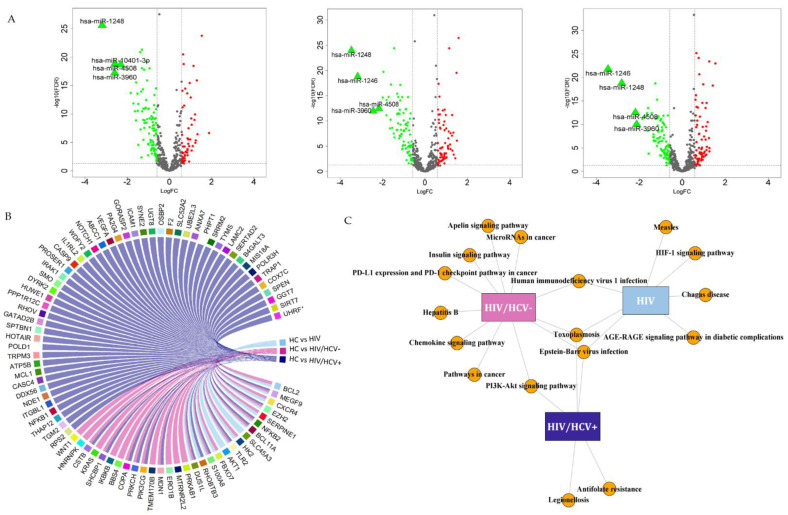
Analysis of differentially expressed miRNAs in HIV patients relative to healthy controls, potential target genes and molecular routes. (**A**) Volcano plots of SDE miRNAs in PBMCs of HIV/HCV+ (**left**), HIV/HCV- (**middle**) and HIV (**right**) patients compared to healthy controls. The expression level change (log_2_FC) and significance (−log_10_FDR) have been positioned on the *x*-axis and *y*-axis, respectively. The colored dots represent significantly differentially expressed miRNAs, indicating upregulation in green and downregulation in red. Those SDE miRNAs with strong up/downregulation (Log_2_FC > 2 or Log_2_FC < −2) are highlighted in wider triangle-shapes and labelled. (**B**) Chord diagram showing the enriched target genes for the SDE miRNAs in the HIV-infected groups (HIV/HCV+ in purple, HIV/HCV- in pink and HIV in blue). (**C**) Network map associating the enriched gene targets and their most significant biological process (KEGG pathways).

**Figure 3 biomedicines-09-01627-f003:**
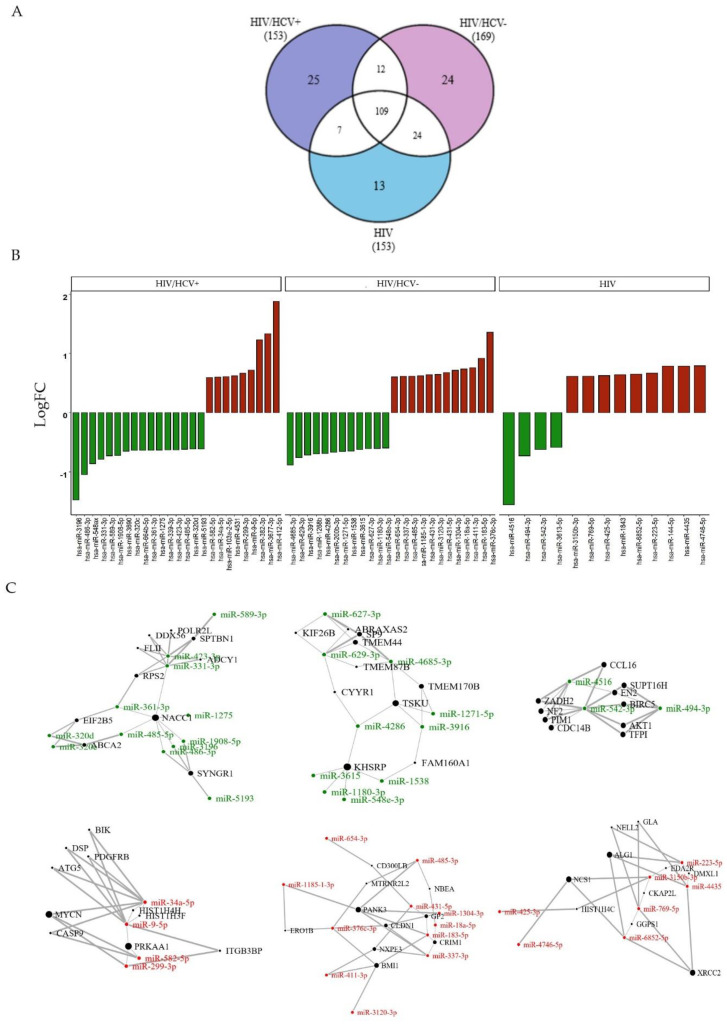
Analysis of specific viral infection signature due to HIV/HCV+, HIV/HCV- or HIV infection: (**A**) Venn diagram representing the distribution of shared and non-shared significant differentially expressed SDE miRNAs among each HIV group (colorless and colored areas, respectively). (**B**) Bar plot of the fold change of non-shared SDE miRNAs (*x*-axis) in the log 2 scale (*y*-axis). Downregulated and upregulated miRNAs are presented as green and red bars, respectively. (**C**) Network graph of the top 10 target genes (black nodes) and their SDE miRNA interactor partners (green/red nodes) for each analysis. Down or upregulated miRNAs in HIV patients have been colored in green and red, respectively. The thicker the line connecting SDE miRNAs to their target gene the more significant the interaction, and the wider the radius the greater number of interactions for each target gene.

**Table 1 biomedicines-09-01627-t001:** Clinical and epidemiological characteristics of the 149 patients enrolled in the study stratified by HCV infection status. Values are expressed as absolute numbers (%) and median (percentile 25 –percentile 75). *p*-values were estimated with nonparametric Kruskal–Wallis test or Man–Whitney U test for continuous variables and Fisher’s exact test for categorical variables: (A) comparison between HC and HIV/HCV+ groups; (B) comparison between HC and HIV/HCV- groups; (C) comparison between the HC group and HIV+ group; (D) comparison between HIV-1 infected groups. Statistically significant differences are presented in bold.

	HC	HIV/HCV+	HIV/HCV-	HIV+	*p*-Values			
** *n* **	32	45	36	36	**A**	**B**	**C**	**D**
**Sex (%)**					0.523	0.830	1.000	0.667
**Male**	16/32 (50.00%)	27/45 (60.00%)	20/36 (55.56%)	18/36 (50.00%)				
**Age (years)**	49.00 (43.75–55.00)	50.00 (45.00–54.00)	52.00 (48.00–56.00)	49.50 (42.75–56.25)	0.592	0.376	0.619	0.250
**Time of HIV Infection (Months)**	-	256.10 (115.12–324.76)	275.70 (142.72–335.41)	219.83 (88.64–265.33)	-	-	-	0.187
**Risk (%)**					-	-	-	**<0.001**
**IDUs**	0/0 (0%)	26/40 (65.00%)	21/31 (67.74%)	0/30 (0.00%)				
**SEXUAL**	0/0 (0%)	14/40 (35.00%)	10/31 (32.26%)	30/30 (100.00%)				
**HIV clinical status (%)**					-	-	-	0.169
**A**	0/0 (0%)	22/43 (51.16%)	13/32 (40.62%)	23/33 (69.70%)				
**B**	0/0 (0%)	7/43 (16.28%)	8/32 (25.00%)	5/33 (15.15%)				
**C**	0/0 (0%)	14/43 (32.56%)	11/32 (34.38%)	5/33 (15.15%)				
**HCV genotype (%)**					-	-	-	-
**1a**	0/0 (0%)	16/39 (41.03%)	0/2 (0.00%)	0/0 (0%)				
**1b**	0/0 (0%)	8/39 (20.51%)	1/2 (50.00%)	0/0 (0%)				
**2**	0/0 (0%)	1/39 (2.56%)	0/2 (0.00%)	0/0 (0%)				
**3**	0/0 (0%)	2/39 (5.13%)	0/2 (0.00%)	0/0 (0%)				
**4**	0/0 (0%)	12/39 (30.77%)	1/2 (50.00%)	0/0 (0%)				
**IL28b (%)**					0.192	0.085	0.730	**0.001**
**CC**	15/29 (51.72%)	14/45 (31.11%)	27/36 (75.00%)	19/36 (52.78%)				
**CT**	12/29 (41.38%)	25/45 (55.56%)	6/36 (16.67%)	16/36 (44.44%)				
**TT**	2/29 (6.90%)	6/45 (13.33%)	3/36 (8.33%)	1/36 (2.78%)				
**CD4+ T cells (cells/mm^3^)**	-	712.00 (530.00–1047.60)	732.70 (560.70–916.65)	833.00 (697.42–1062.00)	-	-	-	0.101
**CD4+ T cells (%)**	-	33.00 (27.00–41.50)	35.00 (31.00–40.00)	38.28 (31.50–44.75)	-	-	-	0.460
**CD8+ T cells (cells/mm^3^)**	-	790.00 (634.00–1048.00)	870.50 (636.50–1104.75)	905.50 (795.42–1244.50)	-	-	-	0.750
**CD8+ T cells (%)**	-	43.68 (12.35)	37.50 (9.40)	38.69 (8.44)	-	-	-	0.158
**CD4/CD8 (%)**	-	0.81 (0.56–1.00)	0.99 (0.75–1.16)	1.06 (0.69–1.27)	-	-	-	0.269

Abbreviations: HIV: Human Immunodeficiency Virus; HCV: Hepatitis C Virus; IDU: Intravenous Drug Users.

**Table 2 biomedicines-09-01627-t002:** Metabolic characteristics of patients enrolled in the study stratified by HCV infection status. Values are expressed as absolute numbers (percentage) and median (percentile 25–percentile 75). *p*-values were estimated by the nonparametric Kruskal–Wallis test and Mann–Whitney U test for continuous variables and by Fisher’s exact test for categorical variables. The following comparisons were performed: (A) HC vs. HIV/HCV+ group; (B) HC vs. HIV/HCV- group; (C) HC group vs. HIV+ group; (D) HIV/HCV+ vs. HIV/HCV- vs. HIV+. Statistically significant differences are shown in bold.

	HC	HIV/HCV+	HIV/HCV-	HIV+	*p*-Value			
** *n* **	32	45	36	36	**A**	**B**	**C**	**D**
**Weight (kg)**	75.80 (64.60–88.00)	64.00 (56.25–73.40)	74.75 (62.62–80.75)	67.00 (61.10–79.00)	**0.001**	0.347	0.173	**0.015**
**BMI**	25.22 (22.78–28.58)	22.48 (20.81–25.72)	24.72 (22.07–27.89)	24.75 (22.44–27.12)	**0.005**	0.500	0.737	**0.026**
**Lipid Profile**								
**Glucose**	90.50 (80.75–93.75)	88.00 (86.00–95.00)	96.50 (89.50–102.25)	92.50 (86.25–96.75)	0.368	0.051	0.282	0.403
**Gluc > 110**	2/26 (7.69%)	3/37 (8.11%)	5/28 (17.86%)	2/26 (7.69%)	1.000	0.480	1.000	0.377
**Total cholesterol**	207.50 (186.50–225.25)	186.00 (166.00–203.00)	192.00 (181.00–221.50)	185.00 (172.50–204.75)	**0.010**	0.178	0.278	0.343
**Chol > 200**	15/26 (57.69%)	10/35 (28.57%)	11/27 (40.74%)	10/26 (38.46%)	**0.043**	0.337	0.267	0.560
**LDL**	-	108.00 (87.00–130.00)	115.00 (100.00–133.00)	115.50 (100.50–144.25)	-	-	-	0.293
**LDL > 130**	-	8/23 (34.78%)	6/15 (40.00%)	9/18 (50.00%)	-	-	-	0.614
**HDL**	67.50 (50.75–78.75)	50.00 (40.00–60.00)	51.00 (45.00–57.00)	46.00 (37.50–54.50)	**<0.001**	**0.002**	**0.001**	0.878
**TG**	85.00 (70.00–134.50)	126.00 (80.00–167.00)	131.00 (118.00–182.00)	116.00 (97.25–178.75)	0.328	**0.038**	0.157	0.535
**TG high**	2/26 (7.69%)	4/35 (11.43%)	6/25 (24.00%)	5/24 (20.83%)	0.960	0.224	0.352	0.412
**LDL/HDL**	-	2.23 (1.66–2.85)	2.42 (1.91–2.95)	2.43 (2.12–2.94)	-	-	-	0.451
**AI**	3.00 (2.58–3.82)	3.83 (3.07–4.42)	4.00 (3.36–4.60)	3.95 (3.67–5.07)	0.085	**0.011**	**0.005**	0.447
**AI low risk**	23/26 (88.46%)	29/35 (82.86%)	17/21 (80.95%)	17/24 (70.83%)	0.806	0.759	0.229	0.519
**AI moderate risk**	3/26 (11.54%)	5/35 (14.29%)	4/21 (19.05%)	6/24 (25.00%)	1.000	0.759	0.385	0.584
**AIP**	0.18 (0.12–0.87)	0.97 (0.32–1.23)	0.92 (0.74–1.44)	0.89 (0.69–1.43)	**0.019**	**0.004**	**0.005**	0.568
**AIP high risk**	12/26 (46.15%)	29/35 (82.86%)	19/21 (90.48%)	23/24 (95.83%)	**0.006**	**0.004**	**<0.001**	0.289
**LCI (×10^3^)**	-	43.33 (27.17–75.66)	70.21 (44.18–81.30)	60.00 (35.84–122.80)	-	-	-	0.731
**Liver Biochemical Parameters**								
**GOT (AST)**	18.50 (15.00–20.00)	37.00 (29.00–46.00)	23.00 (19.00–26.00)	23.50 (20.50–26.75)	**<0.001**	0.055	**0.007**	**<0.001**
**GOT > 40**	0/32 (0.00%)	12/36 (33.33%)	0/28 (0.00%)	2/25 (8.00%)	**0.001**	-	0.366	**0.001**
**GPT (ALT)**	15.00 (12.25–20.75)	45.00 (32.00–55.00)	21.50 (17.00–25.00)	27.50 (22.25–32.25)	**<0.001**	0.675	**0.012**	**<0.001**
**GPT > 40**	2/26 (7.69%)	22/37 (59.46%)	0/28 (0.00%)	4/26 (15.38%)	**<0.001**	0.439	0.664	**<0.001**
**GGT**	19.00 (12.00–28.00)	48.00 (32.25–86.75)	32.00 (26.75–39.50)	26.50 (21.00–41.50)	**0.001**	**<0.001**	**0.014**	** 0.003**
**GGT > 50**	0/25 (0.00%)	17/34 (50.00%)	2/26 (7.69%)	4/24 (16.67%)	**<0.001**	0.488	0.108	**<0.001**
**APRI**	-	0.62 (0.50–0.67)	0.35 (0.30–0.44)	-	-	-	-	-
**FIB-4**	-	1.56 (1.39–1.73)	1.22 (1.18–1.31)	-	-	-	-	**–**
**ALP**	–	85.00 (65.00–96.00)	94.50 (70.25–113.00)	77.00 (66.00–93.75)	-	-	-	**0.038**
**Albumin**	-	4.40 (4.03–4.40)	4.50 (4.43–4.68)	4.15 (4.00–4.60)	-	-	-	0.133
**Hemogram**								
**Vitamin D**	-	22.00 (14.00–27.00)	15.50 (11.50–35.00)	24.00 (21.75–37.00)	-	-	-	0.491
**Calcium (Ca^2+)^**	9.33 (9.21–9.65)	9.40 (9.00–9.70)	9.50 (9.25–9.70)	9.35 (9.10–9.80)	0.297	0.988	0.611	0.414
**Phosphorous (P)**	3.15 (2.60–3.55)	3.30 (2.90–3.60)	3.40 (3.15–3.80)	3.40 (3.15–3.90)	0.351	**0.037**	**0.016**	0.490
**Iron (Fe^2+^)**	86.50 (63.50–119.50)	79.00 (78.50–79.50)	83.00 (65.00–106.00)	-	0.665	0.713	-	-
**Hemoglobin A1c**	-	5.30 (5.30–5.30)	5.60 (5.50–5.80)	-	-	-	-	-
**Leukocytes (×10^3^)**	6.02 (5.28–7.21)	6.91 (5.61–7.65)	7.31 (6.07–8.32)	6.90 (5.71–8.40)	0.313	**0.023**	0.137	0.297
**Red blood cells**	4.90 (4.49–5.12)	4.75 (4.24–5.09)	4.98 (4.90–5.16)	4.83 (4.77–4.96)	0.289	0.506	0.780	0.283
**Hb**	14.15 (12.75–15.45)	15.00 (14.50–15.90)	15.45 (13.67–16.12)	14.90 (13.80–15.67)	**0.008**	**0.018**	0.146	0.376
**Hematocrit (%)**	42.20 (38.62–45.77)	44.70 (43.00–47.50)	46.75 (45.90–47.73)	46.35 (45.63–49.38)	**0.049**	**0.013**	**0.004**	0.363
**Platelets (×10^3^)**	247.00 (206.50–267.50)	219.00 (192.00–249.00)	224.50 (202.00–280.75)	234.50 (220.00–270.50)	0.069	0.962	0.702	0.090
**LHD**	-	175.00 (167.00–195.50)	167.50 (143.00–187.00)	172.00 (162.50–203.50)	-	-	-	0.466

Abbreviations: BMI, body mass index; TC, total cholesterol; LDL, low-density lipoprotein; TG, triglycerides; HDL, high-density lipoprotein; AI, atherogenic index, the cut-off values were: low risk <5% for men and <4.5% for women, moderate risk 5–9 for men and 4.5–7 for women; AIP, atherogenic index for plasma considering high-risk AIP > 0.21; LCI, lipoprotein combine index defined as the ratio of TC*TG*LDL to HDL-C; GOT, glutamate oxaloacetate transaminase; GPT, glutamic-pyruvic transaminase; GGT, gamma-glutamyltransferase; TB, total bilirubin; APRI, AST to platelet ratio index; FIB-4, fibrosis-4 index for liver fibrosis; ALP, alkaline phosphatase; Hemoglobin A1c, hemoglobin glycosylated; LDH, lactate dehydrogenase.

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
