# Peer review of "Different HCV Exposure Drives Specific miRNA Profile in PBMCs of HIV Patients"

_biomedicines, 2021, doi:10.3390/biomedicines9111627_

Round 1

Reviewer 1 Report

The article is well presented and contains a great deal of raw information that is not all fully processed. The supplemental tables present xl worksheets from the instrumental analysis making the content difficult to use. The laboratory data presented (TABLE 2) are not adequately discussed in the dedicated section, making them insignificant in the context of the article. The discussion is thorough and well outlined, but it would be appropriate to comment on the results of a similar article not cited in the bibliography (Moghoofei M, Najafipour S, Mostafaei S, Tavakoli A, Bokharaei-Salim F, Ghorbani S, Javanmard D, Ghaffari H, Monavari SH. MicroRNAs Profiling in HIV, HCV, and HIV/HCV Co-Infected Patients. Curr HIV Res. 2021;19(1):27-34).

Author Response

  1. The supplemental tables present xl worksheets from the instrumental analysis making the content difficult to use.

We used excel format to facilitate data access. However, in order to clarify the use of the information we have included the all data tables in the supplementary material as word tables.

  1. The laboratory data presented (TABLE 2) are not adequately discussed in the dedicated section, making them insignificant in the context of the article.

 Following the suggestions of Reviewer, a new paragraph in which we discuss the data presented in table 2, such as the implications of lipid parameters levels and SDE miRNAs in HIV patients:

-Page 9 lines 128-138: “Analysis of clinic and metabolic characteristics showed significant deregulated levels of lipid parameters of the three groups of HIV patients with respect to HC, while no differences were found within HIV patients. The AIP, which is an index of dyslipidaemia that accurately correlates with the size of HDL-C and LDL-C particles [19], was higher for all HIV groups, especially for HIV/HCV+. Accordingly, we found dysregulation of key lipid metabolism miRNAs such as has-miR-33a, hsa-miR-122-5p, and hsa-miR-125a-5p 20 that were downregulated in all HIV patients. Both, HIV and HCV infections disturb host lipid metabolism generating abnormalities that could lead to severe pathologies [21]. Thus, the miRNA profile of PBMCs reflects common comorbidities in HIV/HCV patients regarding the lipid metabolism, such as an increased cardio-metabolic risk, dyslipidaemia, insulin resistance and hepatic steatosis [22].

  1. The discussion is thorough and well outlined, but it would be appropriate to comment on the results of a similar article not cited in the bibliography (Moghoofei M, Najafipour S, Mostafaei S, Tavakoli A, Bokharaei-Salim F, Ghorbani S, Javanmard D, Ghaffari H, Monavari SH. MicroRNAs Profiling in HIV, HCV, and HIV/HCV Co-Infected Patients. Curr HIV Res. 2021;19(1):27-34).

>We appreciate reviewer comment. The work of Moghoofei et al. has been included in the discussion section:

-Page 9 lines 125-127: To date, only scarce data have been published regarding miRNA profile of HIV/HCV patients [17,18], none of them with a massive approach and even less focused on spontaneous clarifiers

-Page 9 lines 175-178: “In line with our results, Moghoofei et al. 2021 also reported that hsa-miR-122-5p was dif-ferentially expressed in PBMCs of both HIV, HCV and HIV/HCV patients [34]. These out-comes indicate that both HCV and HIV influence in the expression of hsa-miR-122-5p not only in liver cells, but also in PBMCs.”

Reviewer 2 Report

The manuscript by Valle-Miralles and coworkers characterized the miRNA expression profile in PBMCs isolated from patients infected with HIV and HVC. Despite the small size of the clinical universe, the study is relevant, interesting and falls within the scope of the journal.

However, I sincerely think that the work will benefit from a clearer functional analysis. Considering this fact, I would suggest the authors to pay attention to the following points before the article should be accepted for publication.

Point to point comments

1.- In Figures 2 and 3, the color code for the upregulated and downregulated transcripts should be exchanged to follow the classical representation scheme. In this case, please represent in RED the upregulated miRNAs and in GREEN the downregulated ones.

2.- Figure 3 needs to be improved by increasing the size of the legends and letters, specially in panel C.

3.- The functional analysis of the miRNA targets was performed by using a database of experimentally validated targets. This is in principle correct, however limits the conclusions due to the small number of validated targets. In any case, I think that despite of this limitation, the idea of performing an analysis using only validated targets is interesting. However, the functional analysis depicted in Figure 3 is somehow confusing for the reader. I would suggest the authors to perform the same functional analysis but separating the miRNAs that are upregulated from those that are downregulated. So, considering both group of miRNAs, the analysis will be more unbiased and easier to interpret. Both groups of miRNAs should be considered separately, to maximize the selection of regulatory events involving many miRNAs that regulate a common target, and presented as additional results in Figure 3. In biological systems, the most relevant miRNAs should be those upregulated in any of the comparisons. Moreover, the additional results should also deserve an updated discussion.

Author Response

1.- In Figures 2 and 3, the color code for the upregulated and downregulated transcripts should be exchanged to follow the classical representation scheme. In this case, please represent in RED the upregulated miRNAs and in GREEN the downregulated ones.

> Figures 2 and 3 have been updated according to the Reviewer suggestions.

2.- Figure 3 needs to be improved by increasing the size of the legends and letters, specially in panel C.

> As suggested by the reviewer, we have resized and reorganized each panel to provide a more clear view of the results. Additionally, the font size of the labels has been increased.

3.- The functional analysis of the miRNA targets was performed by using a database of experimentally validated targets. This is in principle correct, however limits the conclusions due to the small number of validated targets. In any case, I think that despite of this limitation, the idea of performing an analysis using only validated targets is interesting. However, the functional analysis depicted in Figure 3 is somehow confusing for the reader. I would suggest the authors to perform the same functional analysis but separating the miRNAs that are upregulated from those that are downregulated. So, considering both group of miRNAs, the analysis will be more unbiased and easier to interpret. Both groups of miRNAs should be considered separately, to maximize the selection of regulatory events involving many miRNAs that regulate a common target, and presented as additional results in Figure 3. In biological systems, the most relevant miRNAs should be those upregulated in any of the comparisons. Moreover, the additional results should also deserve an updated discussion.

We appreciate reviewer comments and the suggestion to improve our functional analysis. We have followed his/her guidelines and performed the miRNA functional analysis separating the miRNAs upregulated and downregulated for each comparison. A new figure 3 with this information has been included alongside new paragraphs in results and discussion section as showed below:

-Results section, page 9 lines 93-117: The analysis of miRNA-target gene for each specific group of patients showed us specific interactions which were further analysed by up and down-regulated miRNAs separately (Figure 3B and 3C):

HIV/HCV+ group: we explored the putative target genes for the 25 unique SDE miRNAs. Among the top 10 target genes we found enriched the Spectrin Beta Non-Erythrocytic 1 (SPTBN1) gene, targeted by the downregulated hsa-miR-589-3p, hsa-miR-423-3p, and hsa-miR-331-3p (Supplementary table ST6). The SPTBN1 gene plays a relevant role in biological processes related to the regulation of SMAD proteins and cytoskeleton movement guider. Additionally, miRNA-target analysis revealed that ATG5 (autophagy related 5), PRKAA1 (protein kinase AMP-activated catalytic subunit alpha 1), MYCN, ITGB3BP (integrin subunit beta 3 binding protein), and PDGFRbeta (platelet-derived growth factor receptor beta) might be potential targets of up-regulated hsa-miR-582, hsa-miR-299, hsa-miR-34a, and hsa-miR-9 in HIV/HCV+ patients.

HIV/HCV- group: From the unique 24 SDE miRNAs in HIV/HCV- patients, we found the abraxas 2, BRISC complex subunit (ABRAXAS2) gene, targeted by: down-regulated hsa-miR-629-3p and hsa-miR-627-3p (Supplementary table ST7). Due to its polyubiquitin modification function, ABRAXAS2 plays a key role in protein deubiquitination, cell division and chromosome segregation, among others. Moreover, miRNA-target prediction of up-regulated miRNAs showed significant enriched target genes such as CLDN1 (claudin 1) that could be repressed by hsa-miR-376c-3p, hsa-miR-1304-3p, and hsa-miR-337-3p in PBMCs of HIV/HCV- patients.

HIV+ group: the 13 unique SDE miRNAs showed a panel of 10 genes which interact with downregulated hsa-miR-4516, hsa-miR-542-3p, and hsa-miR-494-3p (Supplementary table ST8). Among them, the ZADH2 (zinc binding alcohol dehydrogenase domain containing 2) encodes for an oxidoreductase that negatively regulate fat cell differentiation.

-Discussion section,

Page 10 lines 184-199:

On the other hand, we explored the specific HIV/HCV+ signature of PBMCs. Functional analysis for up and down-regulated miRNAs were separately performed, to maximize the identification of regulatory events related to phenotypic differences among group of patients [37]. Upregulation of hsa-miR-34a, hsa-miR-9, hsa-miR-299 and hsa-miR-582 suggest that they may be involved in the suppression of key target genes associated with HCV-host interactions as the autophagy related 5 (ATG5), whose inhibition may help to block the HCV replication [38].Up-regulated miRNAs potentially target genes related to liver-disease events such as PRKAA1, MYCN and ITGB3BP [39, 40] . Interestingly, we also observed a potentially target of the platelet derived growth factor receptor beta (PDGFRbeta), highly expressed by hepatic stellate cells (HSCs). The PDGFRbeta is a key mediator in liver injury and fibrogenesis, contributing to human cirrhosis [41]. Therefore, PBMCs likewise HSCs are susceptible to suffer alterations at the transcriptome level in response to HCV infection. Thus, although most miRNAs involved in HCV infection have been characterized in hepatocytes, in this study we showed dysregulated miRNAs in HIV patients chronically infected by HCV involved in liver-related diseases, thereby indicating a similar response between PBMCs and hepatic cells against HCV infection.

Similarly, we also identify specific miRNAs exclusively downregulated in HIV/HCV+ patients…”

Page 11 lines 223-248:

“By exploring the specific footprint of HIV/HCV- patients, we exclusively found 24 SDE miRNAs. Regarding the up-regulated miRNAs, we observed that hsa-miR-376c-3p, hsa-miR-1304-3p and hsa-miR-337-3p are putative regulators of Claudin-1 (CLDN1). This gene encodes for a key cellular co-receptor for HCV entry, which increases susceptibility to HCV infection in the target cells where it is expressed [48]. The up-regulation of the mentioned miRNAs in HIV/HCV- patients will supress CLDN1 expression, and thus, reducing PBMCs susceptibility to HCV infection.

On the other hand, we also analysed those miRNAs specifically downregulated in HIV/HCV- patients. We observed a specific downregulation of hsa-miR-627-3p and hsa-miR-629-3p. These miRNAs are putative targets of ABRAXAS2, which is a part of the deubiquitinating enzyme complex BRISC, whose deficiency resulted in strong attenuation of IFN response [49], which is stimulated by HCV infection [50]. The IFN-signalling pathway has an essential role during the early stages of HCV infection by enhancing and regulating the antiviral immune response [51], where, hundreds of genes are activated. Thus, hsa-miR-627-3p, and hsa-miR-629-3p may be potential biomarkers that highlight molecular alterations specifically due to HCV resolution in HIV-1 patients.

Page 12 lines 270-284:

“Moreover, we explored those specific miRNAs exclusively altered in HIV infection compared to HC. Among the 13 HIV specific up-regulated miRNAs, we found the characteristic miR-223. This microRNA is an important inhibitor of viral replication since it binds to the 3' UTR end of HIV RNA inducing viral latency [13]. Thus, the suppression of this miRNA increases the susceptibility of mononuclear cells to HIV-1 infection [13].

Additionally, we explored the specific down-regulated miRNAs. Our results showed a panel of three downregulated miRNAs (hsa-miR-4516, hsa-miR-494-3p, and hsa-miR-542-3p) involved in the regulation of genes associated with viral latency (SUPT16H, TFPI) [57], HIV reactivation (PIM1) [58], and persistent metabolic and immune disorders (ZADH2, CCL16). Taken all this together, SDE miRNAs in suppressed HIV patients may indicate ongoing abnormalities in PBMCs of HIV suppressed patients, and potential biomarkers for HIV latency. With respect to fatty acid metabolism, we also observed that the down-regulated miRNAs hsa-miR-494-3p, and hsa-miR-4516 also targeted the zinc binding alcohol dehydrogenase domain containing 2 (ZADH2) gene, a negative regulator of fat cell differentiation.

Round 2

Reviewer 2 Report

I am happy with the author's answers and manuscript modifications